# Knickers in a Twist: Confronting Sexual Inequality through Art and Glass

## Sophie Longwill

Independent Researcher, T12 KX00 Cork, Ireland; hello@longwillstudio.com

**Abstract:** Knickers, big, small, plain, sensual, provocative, or practical, can be an unremarkable part of everyday life or an object of feminist protest. Women's clothing, like the experience of womanhood itself, can often have multiple contradictory narratives. In this essay, the author discusses the history of women's underwear and its links with socio-political revolution and feminist art. Against this contextual background, she discusses the development of the body of sculptures entitled *Let's Hook Up*, a series of life-size, paper-thin drawings of lingerie in pâte de verre glass. The author details the artistic processes involved in making the works as well as the conceptual development and exploration of material and meaning. She demonstrates how artwork can act as a gateway to begin conversations about challenging topics like sexual assault whilst also providing a platform for creative expression and connection.

**Keywords:** art glass; diversity; equality; feminism; inclusivity; new narratives; socio-political commentary; studio glass; sustainable development goals (SDGs)

## 1. Introduction

Longwill's work on the series *Let's Hook Up* began in 2016 during the final year of an undergraduate degree in glass at the National College of Art and Design in Dublin, Ireland. (An example of the work can be seen in Figure 1). A referendum was looming to allow the nation to vote on whether to keep the 8th amendment in Irish law which criminalized abortion. As a young woman growing up in Ireland, she had experienced first-hand the grip the Catholic Church still had on the nation when it came to sex education (or lack thereof) in schools, and the shame and stigma instilled from a young age when it came to women's bodies and sexuality. The criminalization of abortion meant an imbedded fear that one might have to 'take a trip to England', if indeed you were fortunate enough to have the means to do so. Longwill had grown up hearing stories about the Magdalene laundries, without fully understanding the atrocities that occurred there. She had witnessed the harsh judgement of teenage girls with whom she went to school who ended up pregnant, whether consensually or not. She had heard stories of mothers pretending their daughter's child was their own to cover up 'the scandal' of it all. As Longwill began to work on this series of sculptures, the artistic process allowed her to unravel the threads between garment history and women's lived experiences, and as she worked with the material of glass, it appeared as a paradigm of womanhood—constantly in flux and full of seemingly impossible contradictions and expectations.

The research methodology for this project was predominantly qualitative. The strategy involved finding examples of women's underwear in different kinds of artwork in order to identify the different narratives that could be told through underwear. This research was carried out extensively using the NCAD library as well as online resources such as JSTOR and the websites of leading contemporary galleries. Primary research involved texts on the history of dress, art and design history with feminist theory, and psychoanalytic theory regarding fetishism to provide a backdrop for the analysis of the artwork. Longwill also studied the history of abortion in Ireland through the archives of national newspapers such as the Irish Times and case studies of the Supreme Court.

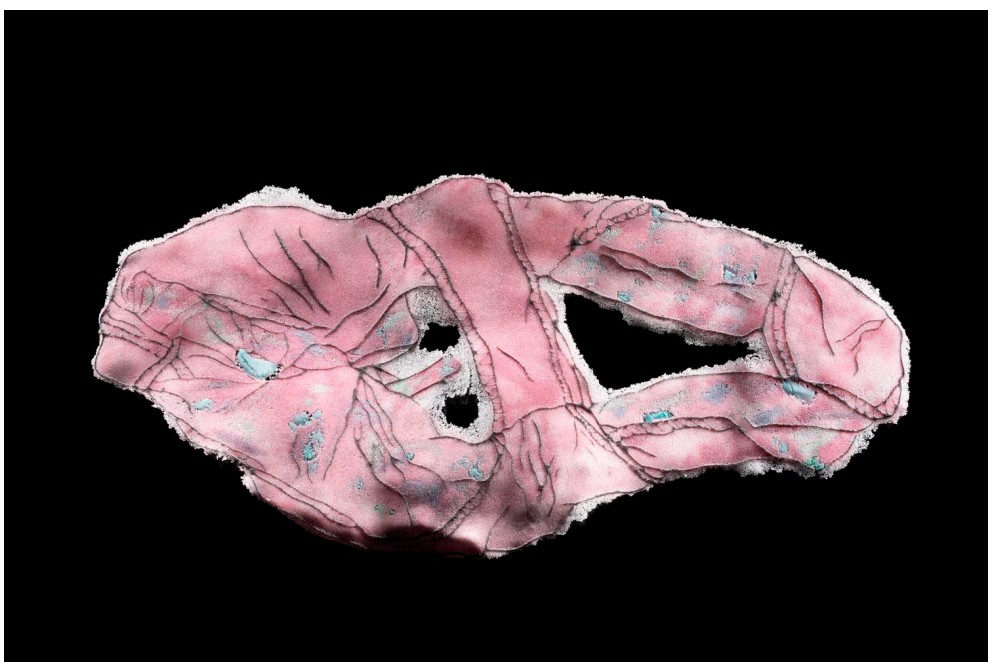

**Figure 1.** Together I, from the series Let's Hook Up, created and auctioned for the Together for Yes campaign. Sophie Longwill. Pâte de verre glass. Dimensions: 50 × 31 × 3.5 cm. 2018. [Created by author].

## 2. Repeal

In 2016, when Longwill began this series of work, protests and rallies to repeal the 8th amendment had been building for several years, particularly after the death of Savita Halappanavar in 2012. The 8th amendment was a subsection introduced to the Constitution of Ireland in 1983 (Irish Statute Book 2023), which recognized the equal right to life of the pregnant woman and the unborn. Abortion was illegal in Ireland and this amendment was deeply problematic. For example, in a number of cases, (e.g., AG (SPUC) v Open Door Counselling Ltd. (1988)), the Supreme Court had held that this provision of the Constitution prohibited information within the state on the availability of abortion services outside of the state (Bailli.org 1992). In the 'X case' a fourteen-year-old girl who had been raped had to have her case appealed to the High Court for an injunction against the Attorney General as she had been restrained from travelling to obtain an abortion in the UK; it was only when she was deemed suicidal that she was granted an abortion to 'safeguard the equal right of the mother' in this case (Bailli.org 1992). Other atrocities occurred to 'protect the life of the unborn', such as in the case of PP v HSE in 2014 where a woman who was clinically brain dead had been kept on life support against the wishes of her family because she was at 15 weeks' gestation at the time of sustaining brain trauma leading to clinical death. This was deeply disturbing and traumatic for the family; it was only when it was finally determined that the chances of the fetus being born alive were 'virtually non-existent' that the family's wishes were honored (The Irish Times 2014).

The tragic death of Savita Halappanavar in 2012 sparked nationwide outcry, which paved the way for the referendum and repeal campaigns. When Savita was 17 weeks pregnant, she was admitted to hospital and it was determined she was having a miscarriage. Her waters broke but the fetus was not expelled, and the following day she requested an abortion with her physician but was denied on legal grounds. She later developed sepsis and died (The New York Times 2018).

Of the ensuing protests and rallies to legislate for abortion rights in Ireland that occurred in light of these tragedies, it was the work of the Speaking of IMELDA collective that really caught Longwill's attention. IMELDA, which stands for Ireland Making England the Legal Destination for Abortion, was established in 2013 and originally used as a code

name by the Irish Women's Abortion Support Group, which supported the thousands of women and girls who traveled from Ireland to England for abortions every year (Speaking of IMELDA 2023). The Speaking of IMELDA collective is a 'feminist intergenerational direct action performance group'. Their work was mischievous and subversive, including creating a campaign called #KnickersforChoice that successfully encouraged people across the country to hang knickers with pro-choice slogans in public spaces and share photos online. They employed humor as a political weapon in their performances to engage the interest of the public and make them feel part of the joke, while the accusation was clearly and firmly placed on the government (Walsh 2020). Longwill was impressed by the sense of community and connection that was created through these campaigns and how this helped to alleviate the sense of shame and stigma that is so ingrained in Irish culture when it comes to women's bodies and sexuality. It sparked her interest in knickers as a feminist object and the long history of underwear in socio-political protest, such as the bra burning of the 1970's and even earlier connections with the Suffragettes and the Bloomers. To contextualize her artistic practice, she began to research the history of the garments to unravel this further.

### 3. History of Knickers

> 'The division between male and female dress has been and remains fundamental to clothing design, and this distinctive apparel has played a central role in constructing gender difference. Clothing thus provides a rich arena in which to explore how masculine and feminine identities take shape and change over time.' (Fields 2007, p. 2)

It was not until the 19th century that the kind of underwear that we are familiar with today began to truly develop, as well as the varied associations and narratives that are imbued within these garments (Saint-Laurent 1968). For many centuries, women's underwear remained very simple in the form of plain shift dresses. It was not until the Middle Ages that clothing of all sorts began to undergo dramatic changes. An interest in the differences between the sexes developed and was reflected in dress, which became increasingly gender specific and elaborate, exaggerating the differences in men's and women's bodies. Men's robes shortened to become tunics and were paired with form-fitting breeches, while women's dresses became longer and more elaborately designed with fitted bodices.

> 'Whilst literature set about exploring the psychological nuances that show men and women are different and not just unequal, dress started to illustrate the physical side of this phenomenon. In short, the Middle Ages added sex to costume.' (Saint-Laurent 1968, p. 66)

This fascination with sexual differences and their representation through dress, particularly underclothes, heralded the beginning of the fetishization of underwear that we know today. In the 19th century, women's underwear became increasingly enmeshed with dualities of meaning, representing a woman's status and sexuality. Tightly laced bodices with whalebone stays accentuated the breasts in an extreme manner for the scopophilic pleasure of others, but at the same time, the tighter the lacing the more it supposedly suggested rigid moralistic reasoning and respectability on the part of the woman wearing it. Particularly in England, there was a belief that a straitlaced woman was not loose (Steele 2001, p. 35), because the constriction of the body was believed to control her physical passions. Rigid stays were emblematic of propriety and respectability and believed to improve the grace and dignity of the wearer. The consensus was that while outer garments could be extravagant and even flirtatious, elaborate and decorative underwear was only for courtesans; respectable women should wear simple and modest undergarments. This overwhelming societal view was epitomized in the words of a French doctor of this period, Doctor Daumas, who claimed, 'The woman's chemise is . . . the white symbol of her modesty, that one must neither touch nor look at too closely' (Steele 1985, p. 193).

At this time, the origins of underwear that we are familiar with today began to truly evolve. Both male and female children of this period wore pantalettes: loose fitting trousers made of either of one or two garments tied at the waist with laces or buttons and an open crotch that was deemed to be more hygienic. Worn under a shorter dress, this allowed more freedom of movement for youthful activities. These garments were also known as 'drawers', as they were drawn onto the body. Drawers soon developed into a type of trousers known as bloomers. In 1851, three leading women's rights activists caused a public uproar when they wore outfits of trousers with knee-length overdresses in Seneca Falls, New York. One of the women, Amelia Jenks Bloomer, was an editor of a women's magazine and, after interest in this new outfit from other women, she published a description of it so others could make their own. Amelia Bloomer became closely associated with this new form of dress although it was not she who invented it; 'Turkish trousers' of a similar style had been worn by fashionable European women and for fancy dress but until then had been deemed too controversial for most American women. The bloomer style soon gained popularity, however, as it allowed so much more freedom of movement and coincided with the beginnings of the movement for equal rights for women. The Bloomer outfit also became known as 'freedom dress' and was adopted by many women's rights activists. The press soon began to denounce the Bloomer costume led by fears that women wearing bifurcated apparel would disrupt the cultural hegemony. The press mocked the Bloomers with satirical articles and cartoons aiming to suppress the dress and freedom related to it.

'Satirical cartoons depicted Bloomer-clad women as crude louts indulging in the worst male vices or bossy wives holding sway over their husbands.' (Callaghan 2010, p. 83). In a way, the satirical approach of the critics worked. Women's rights activists began to abandon the Bloomer outfit as they realized it had become counter-productive; more attention was being paid to their clothing than the contentious issues they were trying to debate. 'A middle-class woman's sexual reputation was often her only source of power in negotiating her material well-being and social standing through marriage. Thus, identifying certain acts as damaging to a woman's reputation was an important and persuasive means of regulating her behaviour, such as constraining her from wearing the Bloomer costume and articulating the women's rights claims it symbolized.' (Fields 2007).

However, the Bloomer outfit did not die out entirely, and the status of women was shifting towards the end of the 19th century. The Suffragettes were working hard to win the right to vote, and the idea of the ideal modern female was no longer that of a 'kept woman' but rather a confident, independent woman who traveled, played sports, and perhaps even drove an automobile. As a result, women needed more freedom of movement and their outfits needed to change accordingly. Corsets were replaced in favor of bras, and hemlines were raised. Women were increasingly breaking free of societal as well as sartorial constraints. They were beginning to enjoy independent living and to have jobs, particularly in department stores. The first department stores were established and grew phenomenally at this time, with Selfridges in London, Au Printemps in Paris, and Neiman Marcus in America. The influence of these stores combined with the development of ready-to-wear, paper patterns, and mail order had a huge effect on fashion at this time and the lives of women (De la Haye and Mendes 2010). Changes in the design and sizing of underwear meant more jobs for women as fitters in department stores but also as designers and creators of new types of lingerie using modern materials. This demonstrates just how closely female underwear has historically been liked to women's freedom and independence.

In the 20th century, drawers were commonly worn by all women and the move to closed-crotch drawers began. With this change in the garment came a new development in associations and moral coding.

> 'The transition from open- to closed-crotch drawers, which was completed by the end of the 1920's, reveals how modesty and eroticism, usually paired as opposites, work together in a shifting dialectic responsive to and productive of changes in understandings about gender difference and sexuality.' (Fields 2007)

As female dress changed and hemlines raised, the closed-crotch drawers that were originally so contentious as provocative bifurcated undergarments became instead encouraged and promoted as favorably modest, offering protection and covering of the genitalia that may have been otherwise exposed under shorter dresses or sportswear. With the outbreak of World War I, it became more acceptable and common practice for women to wear more traditionally male garments like trousers and workwear as they helped in the war effort performing male duties at home while most of the male population was abroad fighting the war. This added to the acceptance of the changing styles of drawers and influenced their design. Drawers became smaller and more form fitting, and they became more commonly referred to as knickers in Europe or panties in the USA.

As women became more active and their clothes and undergarments changed to accommodate this, there came a separation between practical underwear and more sensual undergarments specifically for seduction and sex, i.e., lingerie. This distinction between garments added yet more meanings to female undergarments and what they represented. Advertisements for underwear became more common in magazines and department stores in the 20th century. The way in which female underwear was and still is advertised is key to the formation of the symbolism and narratives associated with it. Advertisers of underwear in the early 20th century had to follow relatively strict moral conventions; although society had progressed greatly in this time and women were more liberated, it was still considered scandalous to show images, whether drawings or photographs, of women in underwear, even for commercial use. Companies resolved this issue by illustrating underwear by itself or with only an illusion to the female form. Often the woman's body was cut off or in shadow. This removal of the female figure really cemented the anthropomorphic quality of underwear, so that the garment represented the woman and stood in for her identity and sexuality. Taking away the female figure did not in fact lessen the erotic appeal of the advertisements and garments themselves, in fact it further accentuated their fetishization as symbols of the female body and sex. It also encouraged the viewer to create their own ideas about the woman behind the garment, and so underwear became emblematic of female identity. Modesty and eroticism have always been closely intertwined and the less is more approach of advertising in this period really demonstrated this (for a full history, see Fields 2007).

Sex has always been related to power for women, especially since for many years the only way a woman could gain power was through seduction of a powerful husband. So lingerie, as a tool of seduction, could be a very powerful item (Fields 2007). The layers of meanings associated with lingerie are, however, undoubtedly complex. These days in Western society, the acquisition of a husband is no longer paramount as women enjoy previously unforeseen levels of freedom and independence. Although lingerie is no doubt worn at times to please a partner, increasingly it is worn to please oneself. Lingerie can make one feel alluring, seductive, and therefore powerful, even if nobody is going to see it. In fact, there can be a certain thrill in this secrecy of intimate garments.

The hyper-sexualization of 21st century advertising for all manner of products means that the lines between commodity and psychological (sexual) fetishism have become increasingly blurred, which in turn has served to reinforce the power of visual representations of lingerie as anthropomorphic symbols layered in meaning. Many feminist artists (examples detailed below) have created satirical works using the language and imagery of advertising to draw attention to the problems inherent in representations of women in visual culture. As the female body and clothing are so ingrained with notions of femininity and idealized beauty, feminist artists often use the body and clothing as tools to subvert these ideas and reclaim the female body from centuries of objectification and the male gaze. Women's underwear is particularly associated with feminism because of its links to the women's rights movement in the times of the Suffragettes and the Bloomers but also to the activities of feminists in the 1970s.

'Although—contrary to urban legend—no bras were burned on the Atlantic City Boardwalk at the 1968 Miss America Pageant protest, participants unfurled the

new critical perspective they called "women's liberation", and feminism has been closely associated with undergarments in the popular imagination ever since.' (Fields 2007, p. 272)

More recently, knickers have been used by the IMELDA group in Ireland in their #knickersforchoice campaign, with knicker bunting and giant knickers installed outside significant buildings around Ireland. This shows how women's bodies as well their lives can be represented by underwear. Underwear can be used in an artistic way to campaign for equal rights and challenge societal norms as well as to create a narrative about the everyday female experience, from the pleasurable to the painful, extraordinary or mundane.

## 4. The Art of Knickers

One of the obvious functions of representations of female underwear in contemporary art is to create narratives relating to sex. As underwear has become a stand in for the female body, it can be used to discuss gender, sexuality, and identity. Underwear can anthropomorphize different materials to reference the objectification of women and derogatory sexual language, as in the case of work by Sarah Lucas (see Figures 2 and 3).

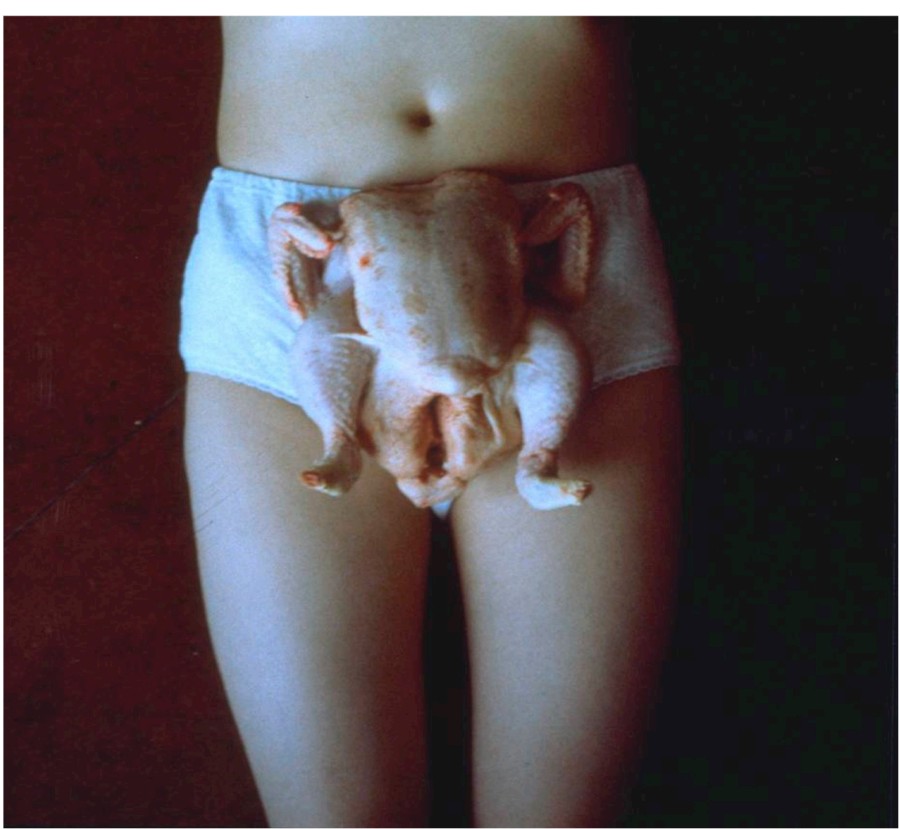

**Figure 2.** Sarah Lucas, Chicken Knickers, 1997. Photograph, color, on paper. Dimensions: 426 × 426 mm.

Other artists may use an obvious lack of underwear and the accentuation of such for shock value and to draw attention to issues of gender in contemporary art, like VALIE EXPORT does (Figures 4 and 5).

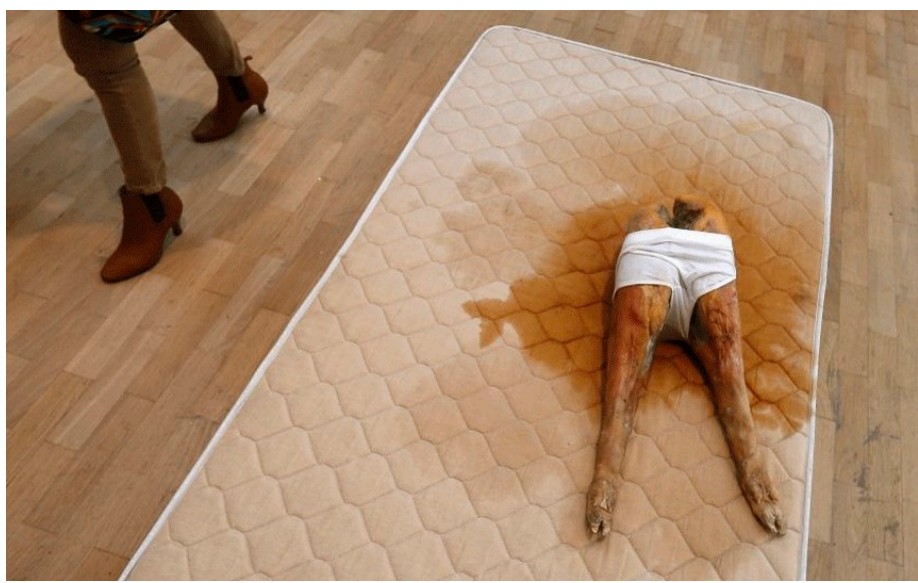

**Figure 3.** Sarah Lucas, I might be shy but I'm still a pig, 2000. Mattress, ham, knickers. Dimensions: 190 × 100 × 39 cm.

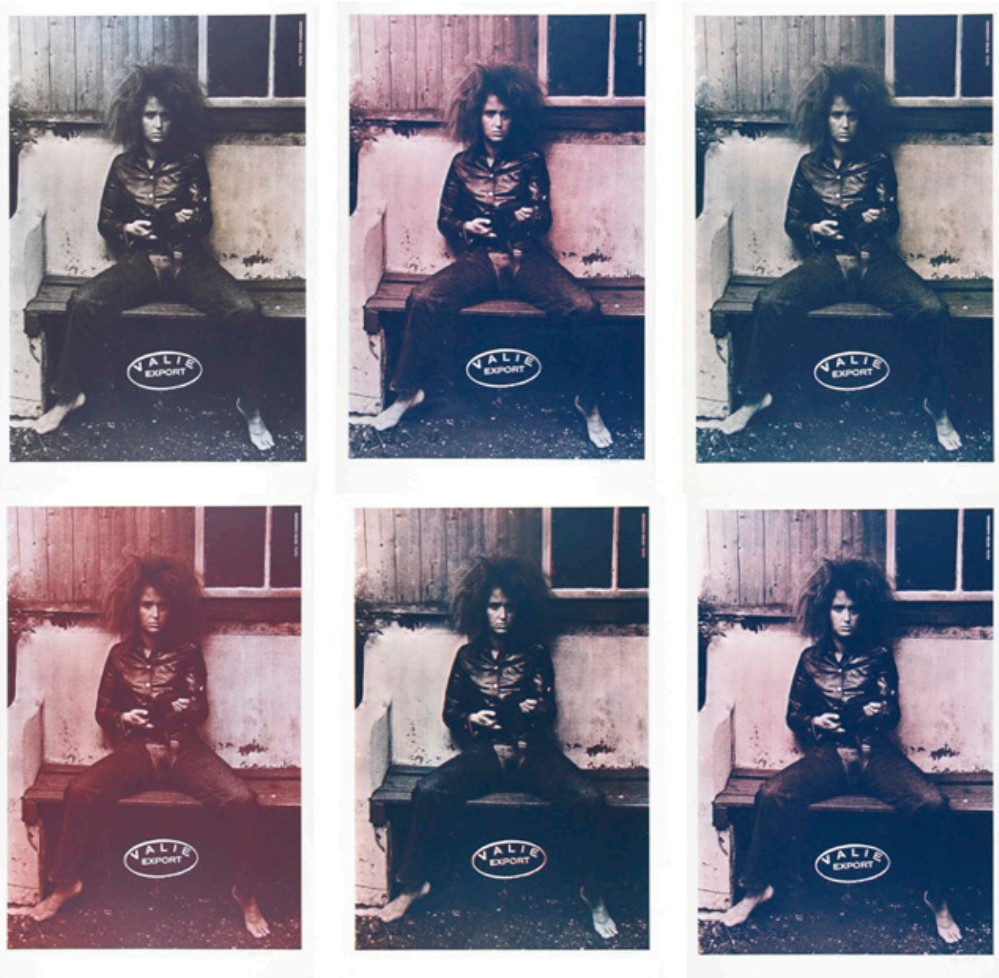

**Figure 4.** VALIE EXPORT, Action Pants: Genital Panic, 1968. Six screenprints on paper. Dimensions: each 658 × 459 mm.

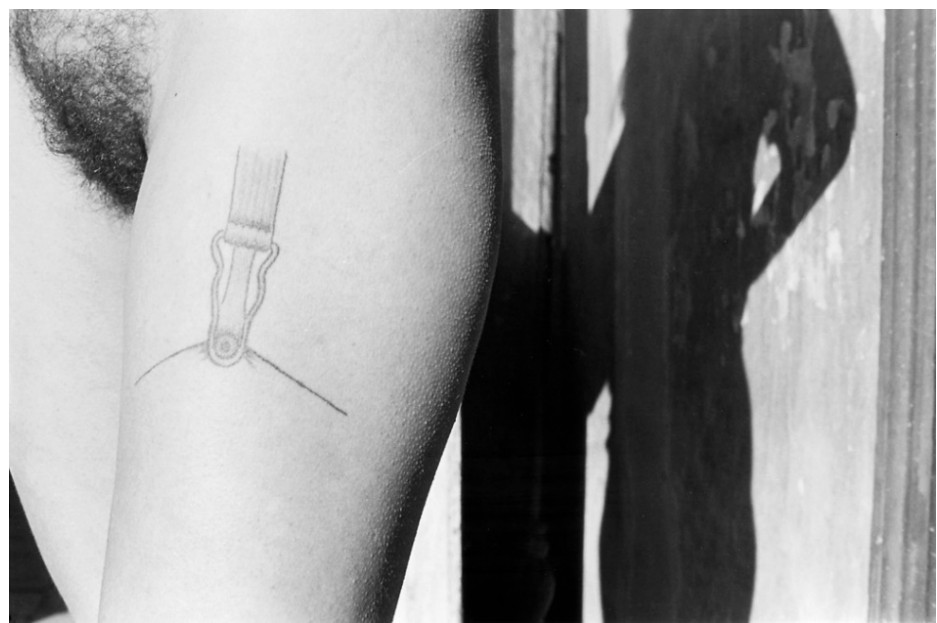

**Figure 5.** VALIE EXPORT, Body Sign Action 2, 1982. Black and white photograph. Dimensions: 100 × 140 cm.

The distinct absence of underwear tells a different story in the work of Scout Pare-Philips, whose warm dreamy photographs draw us in to contemplate the female body and sexuality (Figures 6 and 7).

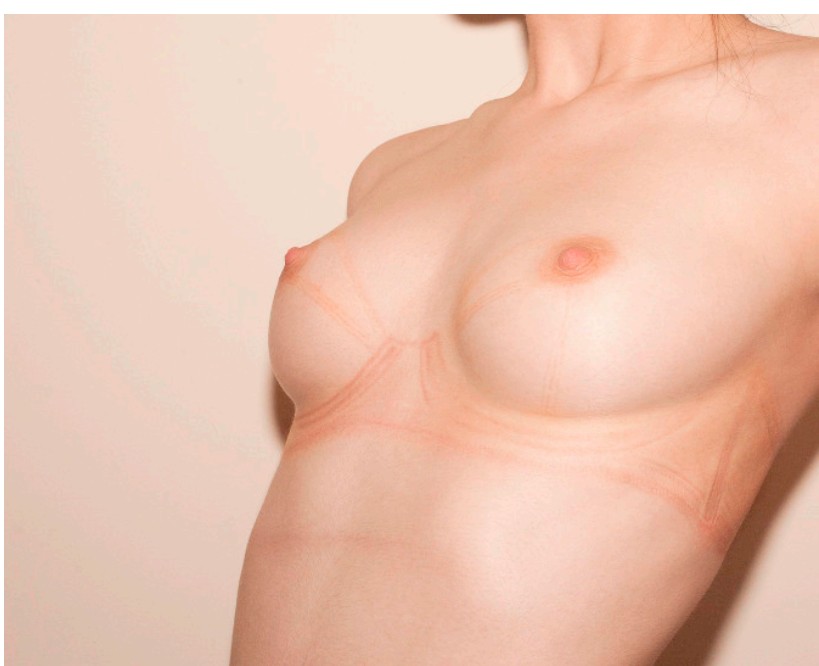

**Figure 6.** Scout Pare-Philips, Impressions, 2011. Photograph, dimensions: unknown.

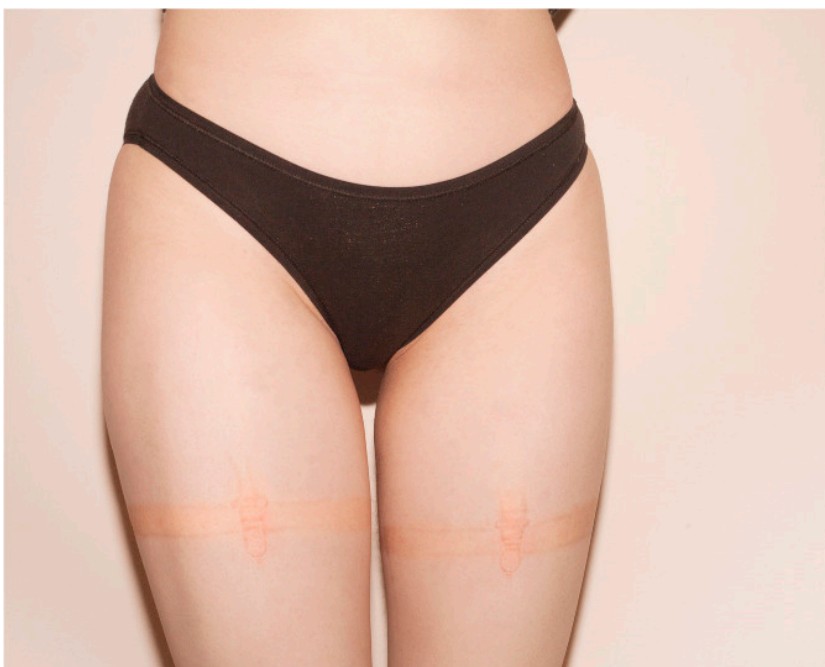

**Figure 7.** Scout Pare-Philips, Impressions, 2011. Photograph, dimensions: unknown.

More blatant reference to sex and fetishism is apparent in the work of Allen Jones. In his life-size sculptures of women bound in submissive positions, it is the type of underwear represented that adds to the narrative (Figure 8). One of the most controversial contemporary artworks involving female underwear is Allen Jones' *Chair*, completed in 1968. Allen Jones is a British Pop artist born in 1937 who became infamous after the exhibition of his pieces in 1970 at the Institute of Contemporary Art in London incited protests. *Chair* is part of a series of three sculptures depicting women as furniture. This piece depicts a life-size model of a woman in rubber underwear, bound into position with a cushion so that she may be sat upon. This piece and the others in the series, *Hatstand* and *Table*, have caused outrage on the numerous times they have been exhibited and have often been vandalized while on show. The sculptures are blatantly fetishist and sado-masochistic; by presenting women as literal furniture, Jones objectifies women in the coarsest way. The combination of rubber and latex fetish underwear and high heels on the female models, as well as their passive expressions and submissive positions, shows that the figures are there to be viewed and used. The viewer is immediately placed in a dominant position over these female figures. The overwhelming misogynist overtones of these works made Jones infamous and shows how a male artist referencing female underwear can use it to create feelings of dominance. Laura Mulvey utilized Freudian psychoanalysis in her critique of Jones' work in *Spare Rib* magazine in 1973. She suggested that the works were a way for the artist to work out his latent castration anxiety because the genitalia of the figures are hidden or absent, so they become phallic forms themselves (Hatt and Klonk 2006, p. 182). Some critics have argued against the readings of misogyny in Jones' work, stating 'Apologists for Jones have argued that his representations of women are merely a lure, drawing the viewer into an aesthetic game' (Sladen 1995). Jones himself claims that he is a feminist; however, he admits himself that is difficult to defend the pieces. 'I can see they are perfect images for an argument about the objectification of women, and if someone thinks that, it is very difficult to gainsay it.' (Wroe 2014). The female underwear in these sculptures adds to the fetishist and scopophilic atmosphere of the work. It also creates a reference point for pornography and hyper-sexualized advertising imagery, so that in this sense it imbues the work with a sense of familiarity for the viewer in that they can engage with and understand the work.

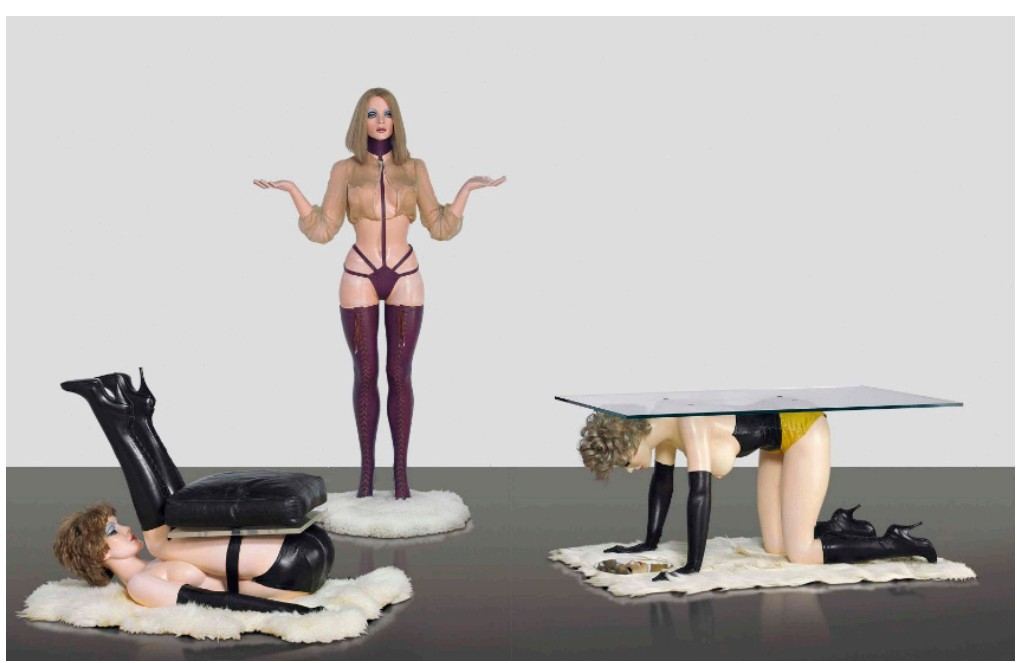

**Figure 8.** Allen Jones, Hatstand, Table and Chair, 1969. Mixed media sculptures: fiberglass, acrylic paint, wigs, Perspex, leather, rubber, latex. Dimensions: variable.

The exploration of female objectification through the use of female underwear in artwork is played out quite differently in the hands of British artist Sarah Lucas (Figures 2 and 3). She explores attitudes towards sexuality and the body by using food and other objects, such as underwear, to represent both male and female body parts, creating pieces that specifically reference slang terms for genitalia. The use of everyday objects by Lucas shows the connections between the body and food, demonstrating how societal attitudes towards sex are perpetuated through language. In the piece *Chicken Knickers*, the artist wears a pair of white knickers with a raw plucked chicken attached so that the orifice of the chicken is positioned approximately over her own vulva. By doing so, she confronts the viewer with a provocative image that represents objectification of the female body.

> 'The presentation of the female genitals as equivalent to a chicken ready to be stuffed and cooked is disturbing and the composition of the image emphasises this narrative. The rest of the artist's body is cut out of the frame so that the age and gender of the model is ambiguous, although the type of underwear worn suggests a young woman or girl. The body is surrounded by an intense darkness which heightens the scene.' (Manchester 2000)

In a later work, *I might be shy but I'm still a pig* (2000) (Figure 3), Lucas uses underwear to hold two haunches of pork, which are displayed on a mattress that is becoming soiled by the meat. The stains from the meat call to mind bodily fluids staining the mattress; they are rust-colored and brown like old blood. The underwear becomes a tool to anthropomorphize and sexualize the otherwise mundane and insignificant pork legs, and combined with the mattress, it creates a distressing image of a violated female form. Lucas' use of food is interesting because it shows how food and sex are so closely linked, and it uses this to give the work a fetishist narrative. The female relationship to food has been documented as being particularly fetishist (see Gamman and Mackinen 1994). Presenting body forms as food creates a narrative of consumption and could read as a dark take on sex and abuse where bodies are expended and then left to decay on dirty mattresses. The humor in Lucas' work comes from the absurd visual of the combination of objects; they appear amusing at first glance and the viewer may laugh in order to protect themselves from the perturbing underlying narrative. This presentation and the wordplay in the titles of the works create an entertaining platform to engage the viewer while enabling the artist to discuss otherwise

difficult subject matter. 'Humour', she (Lucas) has remarked, 'is about negotiating the contradictions thrown up by convention. To a certain extent humour and seriousness are interchangeable. Otherwise it wouldn't be funny. Or devastating.' (Lucas 2015).

The images of work (Figures 4 and 5) by the artist VALIE EXPORT show that the absence of underwear can be just as powerful, if not more so, than the garment itself. The artist created the pseudonym VALIE EXPORT in the 1970's to make an impact for her debut on the art scene in Vienna. That the name is all capitalized is aggressively provocative, a hallmark of EXPORT's work. For the performance piece *Action Pants: Genital Panic*, the artist illustrated her idea of 'expanded cinema' by walking into an art film theater in a leather jacket and crotchless trousers. By exposing her genitalia to the audience in this manner, she forced them to engage with a live woman and performance instead of one on screen. The image of EXPORT in this outfit was taken the following year with the added phallic symbol of a machine gun. The performance was incendiary and a way for the artist to challenge the largely male-dominated art scene in Europe at that time (Marcoci 2010). The fact that the artist was wearing everything but underwear in this performance really drew attention to the genital area and to sexual difference, so it became clear that she was challenging gender disparity in the arts and the inequality of recognition for female artists. There is a strong tradition of female artists using their own bodies in their work. It is an empowering way of subverting the usual objectification of female bodies in art. It is also an effective way of challenging idealized notions of the female body by presenting it in an unusual or provocative way, for example in the work of artists such as Marina Abramovich and Carolee Schneeman. The 1960s and 1970s were a period of high production of provocative feminist art. (For a full history see Phelan and Reckitt 2012). In another work by VALIE EXPORT, *Body Sign Action*, the artist got the image of a garter tattooed on her thigh. This was another way of marking the body as the artwork. That she chose this particular undergarment element added to the provocation of the piece. At that time, tattoos were not as common and generally associated with those on the fringes of society, such as bikers and criminals. It was even more unusual for a woman to be tattooed unless they were part of these underground scenes and the implication was that she might be a prostitute. The garter is a very evocative piece of lingerie. By using her body as a canvas, EXPORT likened it to vellum, which legitimized her use of it (Vanderbeke and Rosenthal 2015). The tattoo is very realistic and placed exactly where a real garter would be if worn. It looks as though the garter is actually pulling up the skin of the leg, which, combined with the painful process of tattooing, heightens the level of discomfort for the viewer. This works perfectly for the artist, as she called garters 'a symbol of former enslavement' . . . 'the artist declares garters as a special piece of sexy underwear as outdated as the gender norms they represent which reduce women to sexual objects' (Vanderbeke and Rosenthal 2015). Again, we see how a reference to female underwear can charge an artwork with layers of meaning.

Another artist dealing with female underwear through its marked absence is Baltimore artist Scout Pare-Phillips. In the series *Impressions*, the marks left from the wearing of lingerie are presented in photographs of the imprinted body (see Figures 6 and 7). The faint lines left on the body from the underwear could be read as being like the marks left from a lover. It is at once both beautiful and unsettling. The surrounding parts of the model's body are cut off. The female body is not usually presented in this way with marks, as unblemished skin is so idealized in visual culture. The lack of other marks on the body accentuates the imprints, and combined with the inviting warm tones of the photographs, they certainly engage the viewer. The marks from the underwear showing the restriction of the female form could serve as a reference to wider issues of female restriction. Although subtle, the marks on the body appear almost as brands or burns, as if the woman has been forced to wear something uncomfortable. This could, like Susan Taylor Glasgow's piece *Eve's Penance* (see below, Figure 9), be a reference to the idea of suffering for beauty. Throughout history, women have been derided for this particular perceived vanity, especially in relation to corsetry. 'The tyranny of the corset was always

present. Modern feminists see it as a garment of shame, shackling women for the pleasure of man, but it is apparent that much of the time it was a shared pleasure.' (McDowell 1992, p. 29).

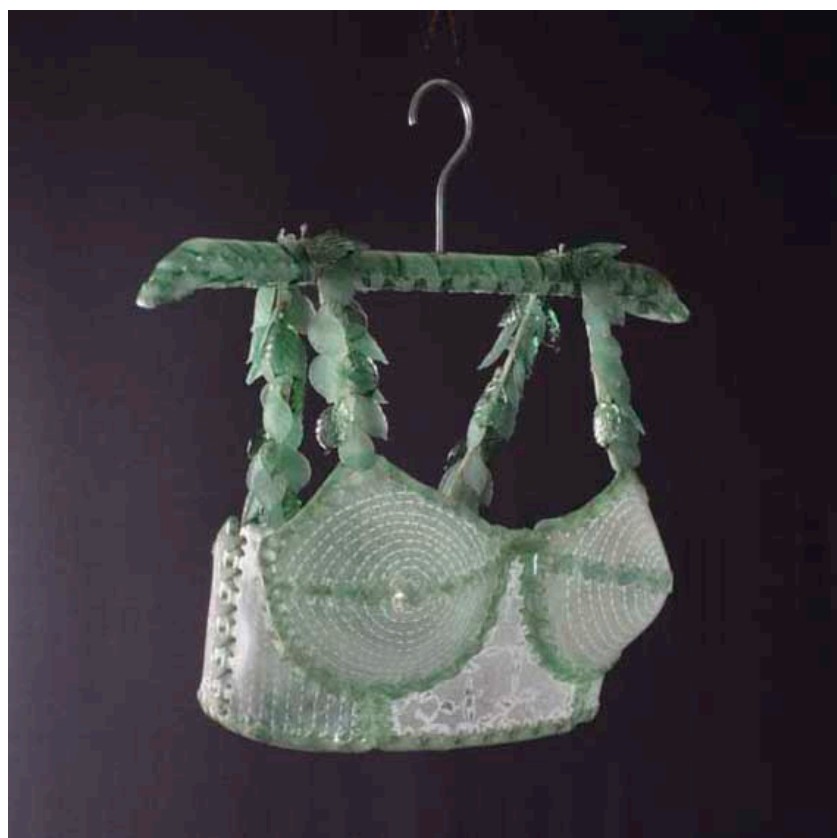

**Figure 9.** Susan Taylor Glasgow, Eve's Penance, 2008. Fused, slumped, enameled, torch-worked, sewn glass. Dimensions: 18 × 14 × 10 inches.

### 5. Material and Meaning

How does material affect meaning? This section will examine works by artists that specifically use the medium of glass. It is an unusual material with its own connotations. Glass can be manipulated in endless ways, so the processes and types of glass used are also important to these pieces. By analyzing the functions of female underwear in the works of these artists, this chapter will demonstrate the added layer of narrative that use of a particular material can add to an artwork. One of the functions of female underwear in contemporary art is to provide a platform to discuss traditional female roles. As underwear is an everyday object that represents women, it is the perfect item to use in work exploring the female experience and women's daily lives, what is expected of women, and how that can make one feel. By using underwear in an unusual way, female artists can challenge societal notions about women and gain a better understanding of their own identity. It is also a way of engaging viewers to think about difficult subject matter through the safer platform of visually pleasing sculptures. This is evident in the work of Susan Taylor Glasgow, an American glass artist based in Colombia, Missouri. She looks at the different tasks that women are expected to perform to be considered a 'good wife', such as baking cakes and sewing dresses, and then subverts these by making the items in glass so that the cakes become inedible and the clothes unwearable. She has a background in dressmaking and the way she uses glass is unusual and interesting as it is clearly influenced by the methods she would have used as a seamstress. She cuts panels of glass and assembles them in a pattern as one would with cloth. She then slumps the panels in a kiln so that the rigid sheets of glass flow into the desired form and acquire some of the nuances of fluidity

that cloth has. She then stitches all the panels together with thread. The process is very time consuming and reflects the time-consuming nature and indeed tedium of traditional domestic work. When she creates female underwear in glass, she makes bras and corsets, which are the most restrictive garments. The rigidity of the glass accentuates the feeling of constriction. However, by using glass as the material, there is also the sense that these restrictive garments could easily be shattered, creating an underlying narrative of female emancipation. Her work shows how significantly material can affect meaning. The piece pictured in Figure 9, titled *Eve's Penance*, could seem to suggest that wearing restrictive and uncomfortable undergarments is a way of making up for the original fall of man or is a way of atoning for our supposed sins through self-punishment, similar to the self-flagellation practices of monks in older times.

The reference to Eve and women's association with nature is alluded to again in her later work *Spring Garden Chandelier* (see Figure 10) from her series of chandelier dresses. As female underwear represents the female body, incorporating the glass corset into a lighting fixture in this way clearly reflects the objectification of women and their domestic duties. The artist seems to say that women are expected to literally light up the home as domestic goddesses. Susan Taylor Glasgow says that 'she 'sews, cooks and arranges' glass to reconcile the conflict over work and home, feminist ideals and the Madonna complex, duty and fulfillment. The bustier forms of her Chandelier Dresses are a fantasy, which reminds us of the way things never were.' (Heller 2023).

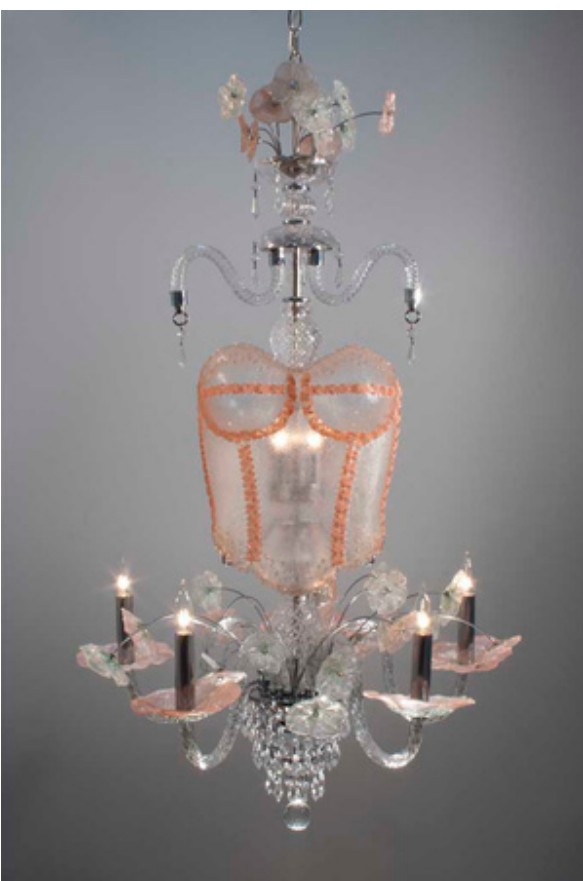

**Figure 10.** Susan Taylor Glasgow, Spring Garden Chandelier Dress, 2014. Glass/mixed media. Dimensions: 42 × 24 × 24 inches.

Liza Lou is an American artist who has worked with tiny glass beads to make life-size sculptures. The piece pictured in Figure 11 is one of her early works. At the beginning of her career, she focused on recreating domestic items in realistic bead sculptures. Her most well-known piece from this period is *Kitchen* (1991–1996), which is a full-scale reproduction of a

kitchen with each item meticulously handcrafted with minute beads. This piece brought her career to prominence when it was shown at the New Museum of Contemporary Art in New York in 1996. Her work explores themes of materiality, labor, and confinement, with many of her later works relating to prisons. The detail in works such as *Yellow Panties* (Figure 11) is truly remarkable and the process is incredibly labor intensive. By using domestic items, the artist references the drudgery of housework and how restrictive it can be. By using glass beads instead of sheets, Lou creates dynamic movement in the work that belies the rigidity of the material. Her early works have a distinctive pop art feel and engage the viewer with the use of bright colors, drawing them in to look closer and truly appreciate the fine detail of the sculptures. The way in which Lou has represented female underwear is interesting as it appears playful on initial viewing. The bright colors and patterns in the pieces as well as the flowing lines of the form suggest fluidity and movement, even though the glass beads are very hard. She has recreated the sense of cloth very successfully, but by using a contrasting material, the result is a little unsettling. It looks almost as though the underwear has just been worn and then discarded, which invites readings of sexual undertones. Taking an everyday item such as underwear or indeed other domestic objects as she does, and recreating them so painstakingly with beads, makes the items transcend their usual mundanity. Like Susan Taylor Glasgow's work, by making unwearable female underwear in glass, the artist creates a narrative of restriction and the time-consuming tedium of traditional female domestic roles. *Yellow Panties* and other items from the underwear and socks series were made by Lou at the same time as she was working on *Kitchen*. With these pieces, she took used underwear and socks and covered them in plaster and beads, transforming the discarded into something magnificent. This transformation makes the underwear more closely linked to the original meaning of the word fetish, i.e., a magical object. Lou's work shows perfectly how female underwear can have many contradictory narratives; it can be discarded, mundane, used, and unpleasant, but also beautiful, pleasing, and valued. In the case of *Yellow Panties*, it encapsulates all of these at once.

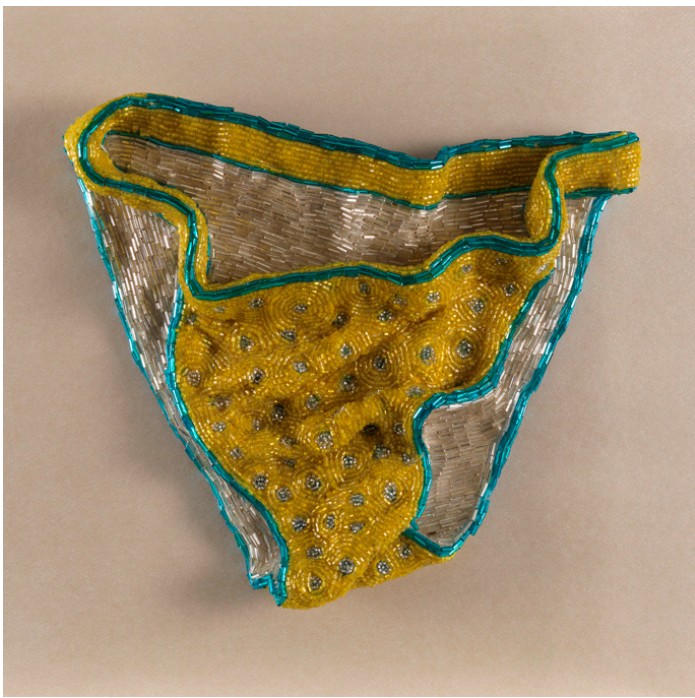

**Figure 11.** Liza Lou, Yellow Panties with Tiny Light Blue Polka Dots, 1995. Beads, papier mâché. Dimensions: $9\frac{3}{4} \times 10\frac{3}{4} \times 1\frac{1}{2}$ inches.

The theme of the discarded and overlooked everyday object is also apparent in Wang Zhiyuan's work using female underwear. Wang Zhiyuan is a Chinese artist who works with refuse and re-orders it to create impressive towers or combines it with handmade elements in the form of everyday objects on a giant scale, such as in the piece *Purge*, pictured in Figure 12. The artist said about this piece, 'The focal point of this artwork is on the hidden ecologic and social crisis behind the present industrial civilization, thus it explains the lifestyle of electronic modern age by comparing the "discarded" to the "excreted". I believe that this electrical waste is not only the entity of waste itself but contains as well, the message junk relays through its dissemination.' (Wang 2023). The underwear in *Purge* could be male underwear, but the artist's previous works and the way this is presented suggest that it is female underwear. The rubbish spews forth from the labial folds at the front of the piece as if being birthed, suggesting that all we produce is rubbish. Zhiyuan has created many pieces recreating female underwear on a large scale in fiberglass. The first series, *Underpants*, which he made in 2001, were direct copies of real underwear, and he then expanded by designing his own in 2003–2005 and began combining different elements like neon lights. He creates underwear sculptures on a large scale predominantly to explore the effect of putting underwear where it does not belong, such as on the wall of a gallery. As his pieces have developed, they have become a way of exploring issues of sex, religion, the AIDS crisis, censorship, and more.

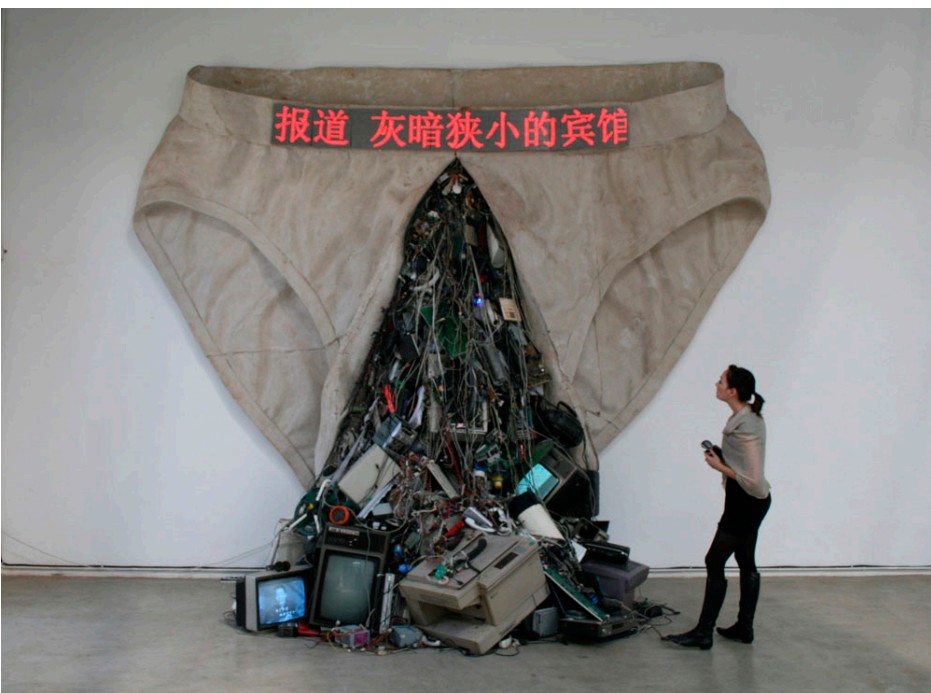

**Figure 12.** Wang Zhiyuan, Purge, 2009. "A gray and narrow hotel." Fiberglass, electronic rubbish, sound, and LED. Dimensions: 510 × 413 × 230 cm.

His piece *Object of Desire* (Figure 13) depicts a bar scene with a devil figure and neon lights that read 'Diamonds Matter Most'. It reads as a depiction of all that is 'sinful': sex, partying, and materialism. This is emphasized by the song that accompanies it 'When are you coming again' by Zhou Xuan, which was banned in China until 1949 for being too decadent. It seems the artist wants to draw attention to these activities that may be overlooked in the same way as discarded underwear. By creating large-scale sculptures of underwear, Wang Zhiyuan creates a motif that draws attention to the narratives he wants to discuss. In this way, the shock value of underwear in an unexpected setting and presentation can function as a platform to engage audiences. Using fiberglass as the

material means the sculptures are closely detailed and very realistic in shape and form compared to the original garment.

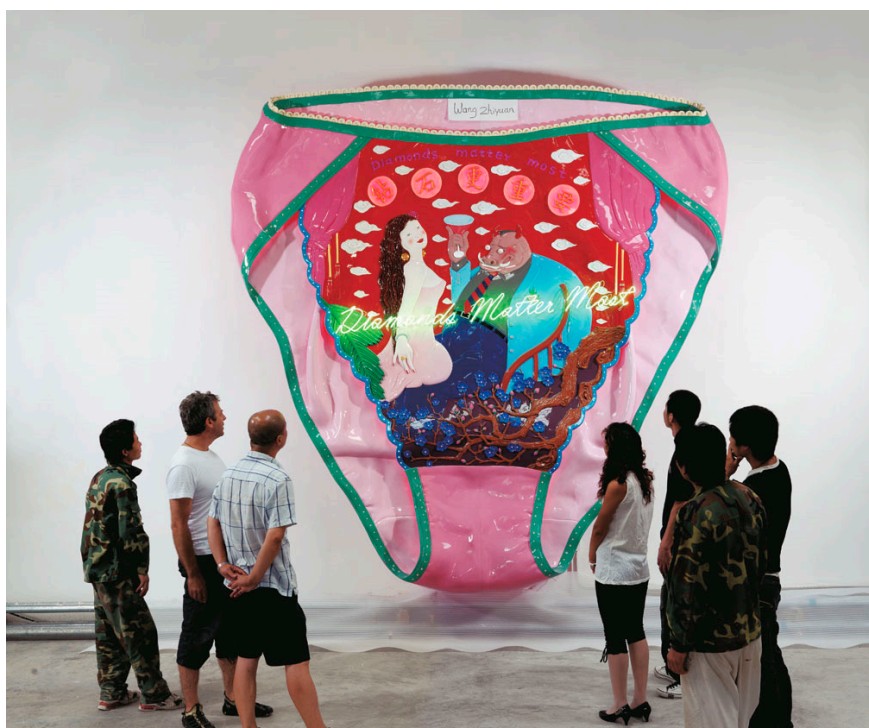

**Figure 13.** Wang Zhiyuan, Object of Desire, 2008. Fiberglass, baking paint, light, sound. Dimensions: 355 × 363 × 70 cm.

The blown glass knickers by American artist Katie Plunkard (Figure 14) use glass in quite a different way to the previous artists. By hot-sculpting glass once it has been blown, the pieces acquire a lot of movement and playfulness, which is highlighted by the use of bright colors. It is much harder to obtain precise realistic details on hot glass as it is constantly moving and difficult to work with, so using this method produces pieces that are quite cartoonish. The combination of this method with the bright colors and added sculpted objects gives these pieces a pop art feel. It makes them engaging and humorous. The addition of elements such as the clock, keyhole, crab, and grenade makes it clear that these pieces represent sexual and political issues. Addressing the work, the artist stated, 'Underwear is traditionally imbued with a variety of meanings . . . I see the series addressing a number of topics including sexuality, gender, fetish, health, politics, history, social norms and more, I appreciate that viewers will likely approach the work from different perspectives' (Plunkard 2015). The artist acknowledges that many viewers may not delve deeper than the initial humor of the pieces but equates this engagement as satisfactory. If the pieces were to be cast directly from underwear, they would pick up a lot more detail and appear very realistic but would lose the playfulness and dynamic movement achieved through the use of hot glass. The function of female underwear in these works is to reference difficult topics but also to engage viewers through humor and fun.

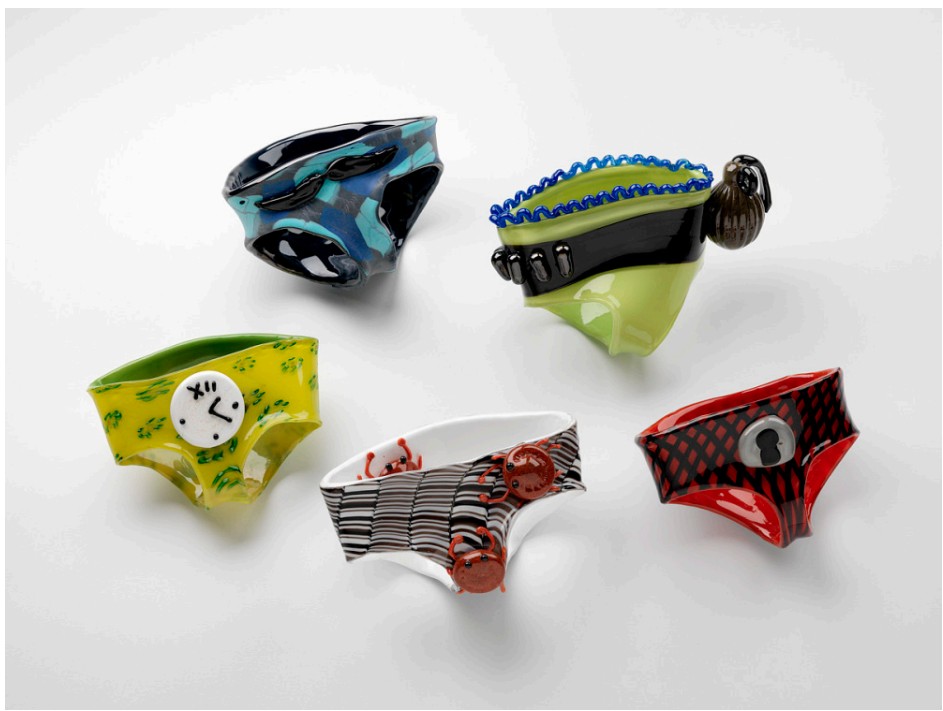

**Figure 14.** Katie Plunkard, Unmentionables, 2011. Blown and sculpted glass. Dimensions: each approx. 15 × 10 × 10 cm.

All of these artists, while using the same material, have created many different stories through their pieces. Glass is imbued with its own meanings, many of which are contradictory depending on the context, just like underwear. It can be pure, stained, hard, fluid, transparent, opaque, colorful, strong, or shattered. Using glass as an artistic material may make people think of traditional stained glass, everyday glass homeware, or cutting-edge glass technology in smart phones and spacesuits. As it has so many of its own conflicting associations, it is the perfect medium to explore the ambiguities and symbolism of lingerie and to add extra layers of meaning to the narratives of these works.

## 6. Let's Hook Up

Alongside this research, Longwill commenced her own artistic response. She began with drawings, paintings, papercuts, and prints and explored artistic responses to underwear and lace. She was interested in the oxymoronic narrative of a garment that in many ways is an unremarkable part of everyday life but also potentially very provocative and controversial. She was fascinated with how the material and process used could alter the underlying narrative and the interpretation of the work. Following her research and much experimentation, the artist determined glass to be the perfect medium to explore the narratives of lingerie that fascinate her: the separation from functionality to highly decorative intimate apparel that is sensual and often impractical creates layers of meaning that are often confusing and contradictory. Glass is a seductive substance with a contradictory nature—both fragile and dangerous. It can be both solid and liquid, hard and soft, transparent and opaque. It seemed that the material itself with all its conflicting narratives and expectations could be viewed as a paradigm for womanhood. It has an emotive quality that accentuates the powerful underlying messages presented in its artistic use (Figure 15).

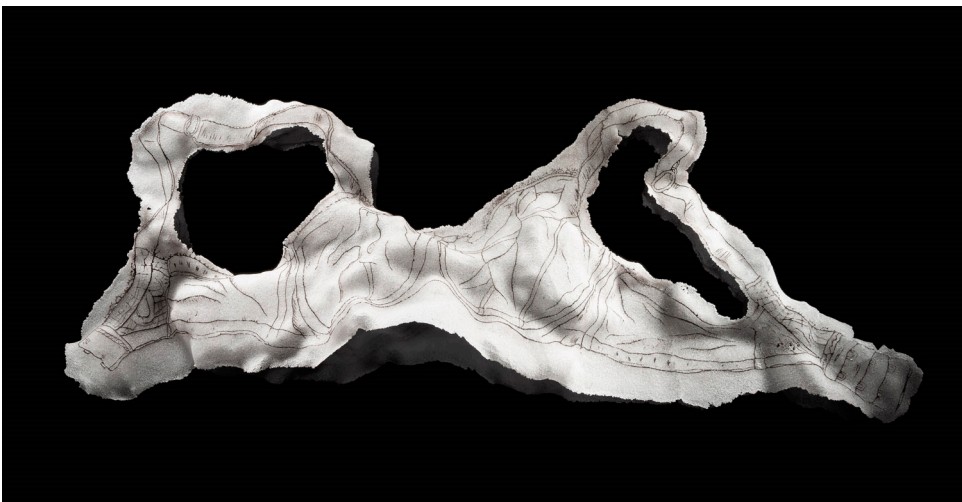

**Figure 15.** Fallen Bra, from the series Let's Hook Up, Sophie Longwill. Pâte de verre glass, hot sculpted and slumped, 2019. Dimensions: 64 × 24 × 10 cm [Created by author].

Longwill experimented with different processes to represent lingerie with the material. She tried printing and painting imagery in the style of traditional stained glass. She sculpted blown glass knickers in the hot shop, but they had a quality that felt too playful for the underlying context (see Figure 16). She cast directly from knickers themselves in multiple ways with wax and fabric to create solid glass sculptures of discarded underwear, or the impression of a pair, but although the beauty of the details of the folded fabric and lace was engaging, it felt far too direct.

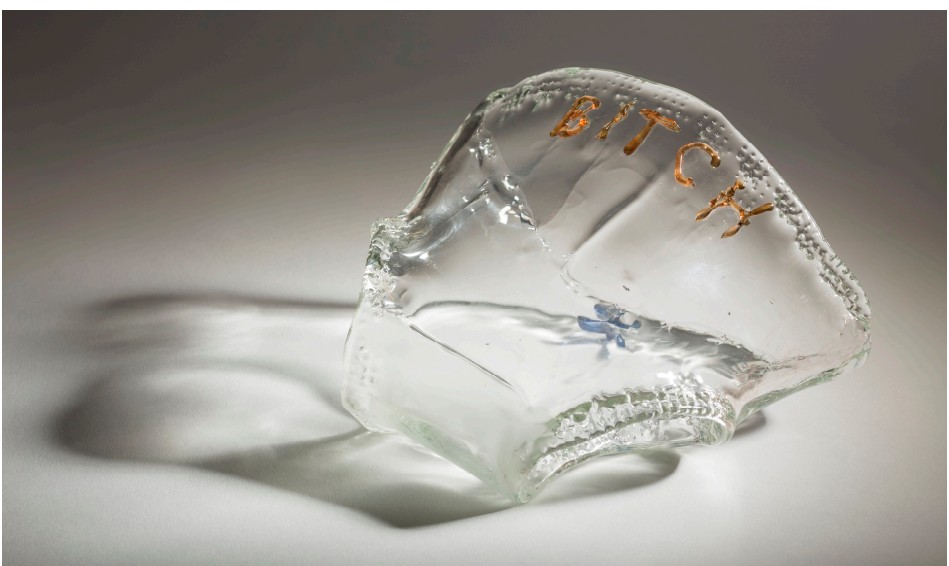

**Figure 16.** Bitchpants, from the series Let's Hook Up, Sophie Longwill. Blown Glass, 2019. Dimensions: 16 × 12 × 9 cm [Created by author].

Finally, the artist discovered the technique of pâte de verre and a way of translating into glass the watercolor paintings that she had been creating as she documented stories throughout the process (Figure 17). Longwill did not want the pieces to be direct renditions of the underwear; it was important to the artist to give a sense of having captured a fleeting impression and the emotion of that moment. The watercolors captured a sense of this, which became amplified through the use of glass. The pieces are drawings of underwear that has just been taken off (see Figure 17). There is no explanation of where, when, or why,

just a pervading glimmer of vulnerable beauty that is accentuated by the fragility of the glass.

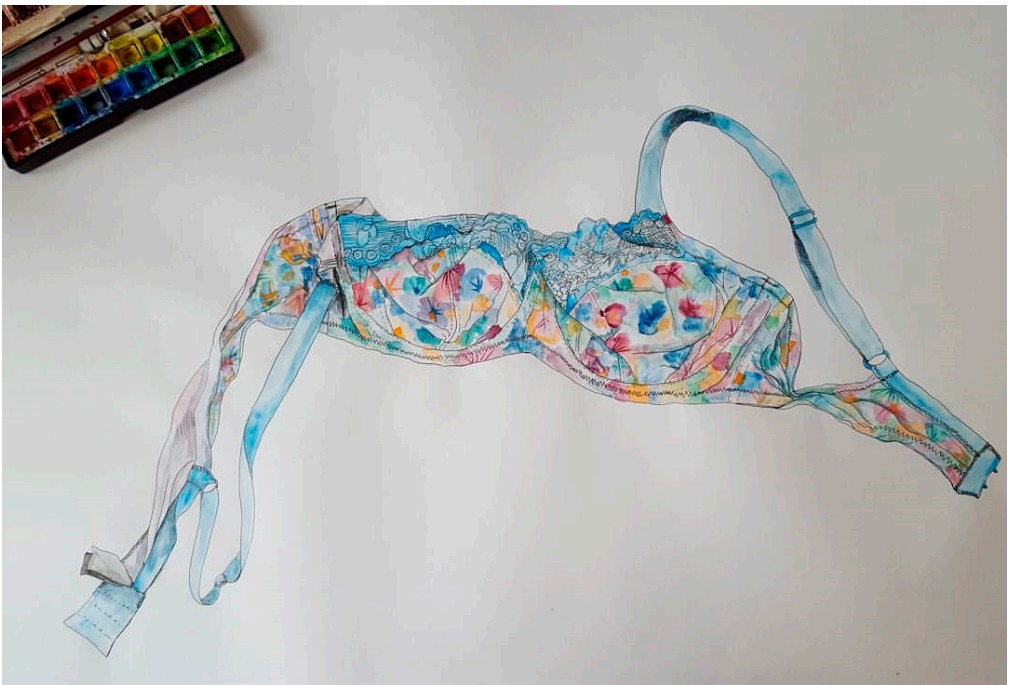

**Figure 17.** Preparatory drawing for glass sculpture in Let's Hook Up. 2016. Watercolor on paper. Dimensions: 84 × 58 cm. [Created by author].

Longwill created the sculptures by making large, flat, plaster/silica molds into which she carved drawings (Figure 18). With a background in printmaking, she treated the molds like intaglio plates. Instead of ink, she pushed finely ground glass powder into the carved lines, which, upon firing, would become raised upon the surface of the glass like the delicate lines of stitching. Once the outlines were complete, she mixed colored glass powders into a paste with glue and water and built them up in layers to replicate the colors and overlapping patterns in textiles (Figure 19). Longwill kept the glass purposefully thin and delicate, almost intimidating to touch, to give a sense of danger and delicate beauty. For this reason, she also kept the natural 'dragon's teeth' edge that occurred during firing as the glass pulled together and fused into sharp points along the edge (Figure 20).

Having initially displayed the works in this series as framed flat pieces, Longwill continued to develop the work and added more movement and flow to the pieces through slumping and hot sculpting, while also experimenting with their display in exhibition settings to make the work increasingly dynamic and engaging (see Figure 21). For example, in the exhibition I See Crimson, I See Red (2019), the works were displayed in the domestic setting of an abandoned house in Elizabeth Fort, Cork, and were reviewed by political scientist Chiara Bonfiglioli for *ArtLeaks Gazette* #5: *Patriarchy Over & Out. Discourse Made Manifest* (Bonfiglioli 2019).

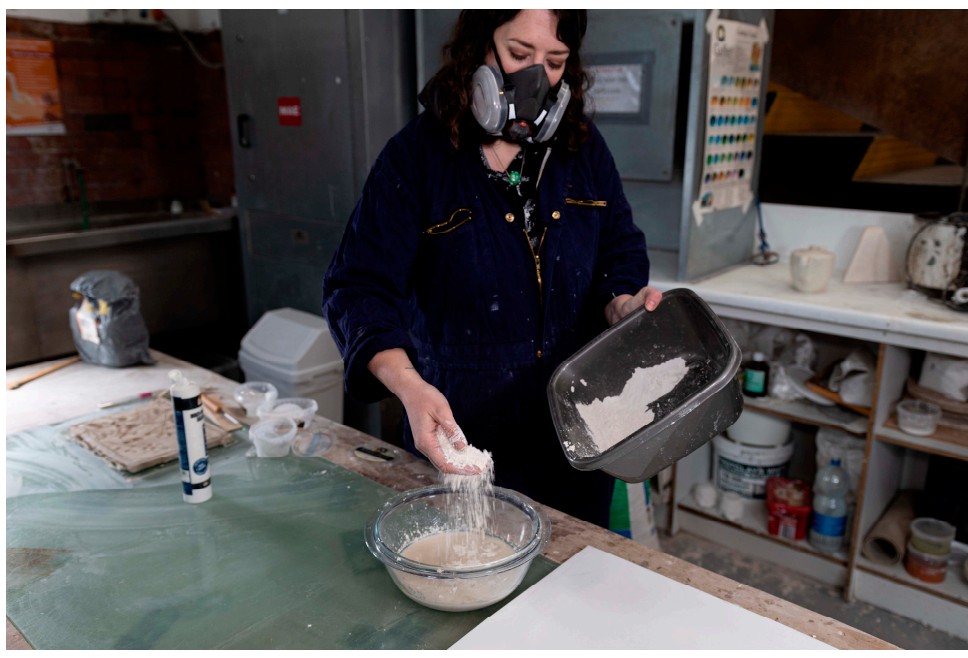

**Figure 18.** Longwill creating plaster/silica molds for glass. Photo by Joe Lington.

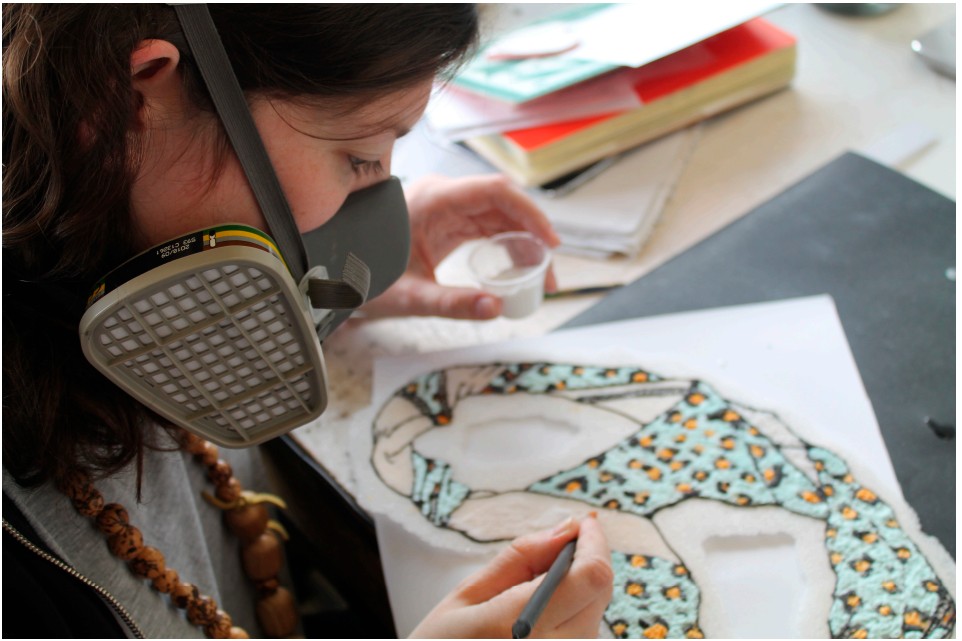

**Figure 19.** Longwill working with glass powders to create a pâte de verre glass sculpture. Photo by Alice Power.

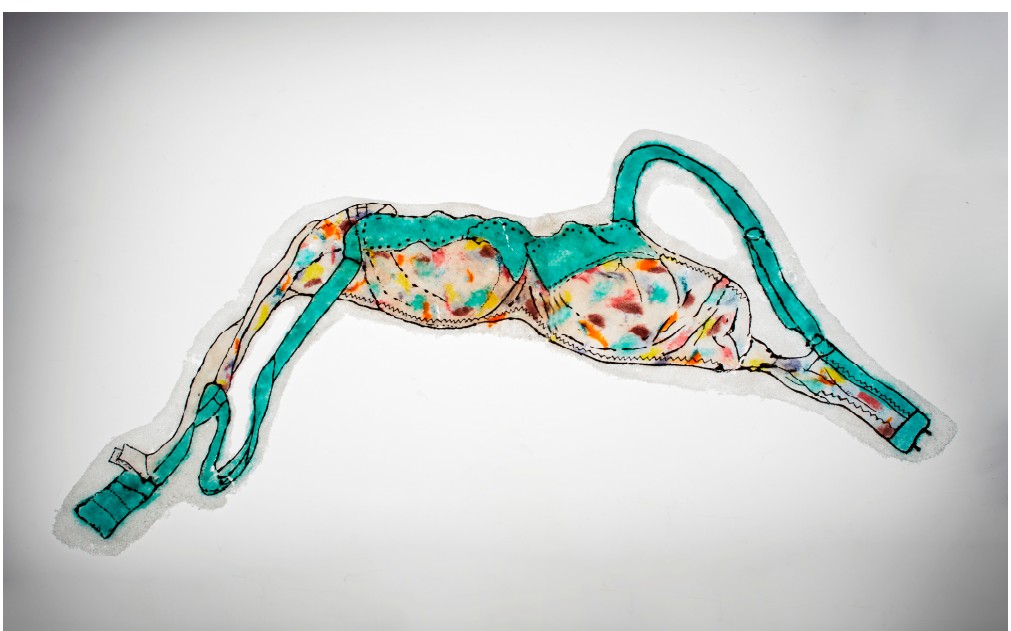

**Figure 20.** Let's Hook Up, glass sculpture from the series Let's Hook Up, Sophie Longwill. Pâte de verre glass. 2016. Dimensions: 83 × 58 × 3 cm. In the collection of the Dox Centre for Contemporary Art, Czech Republic. 2019. [Created by author].

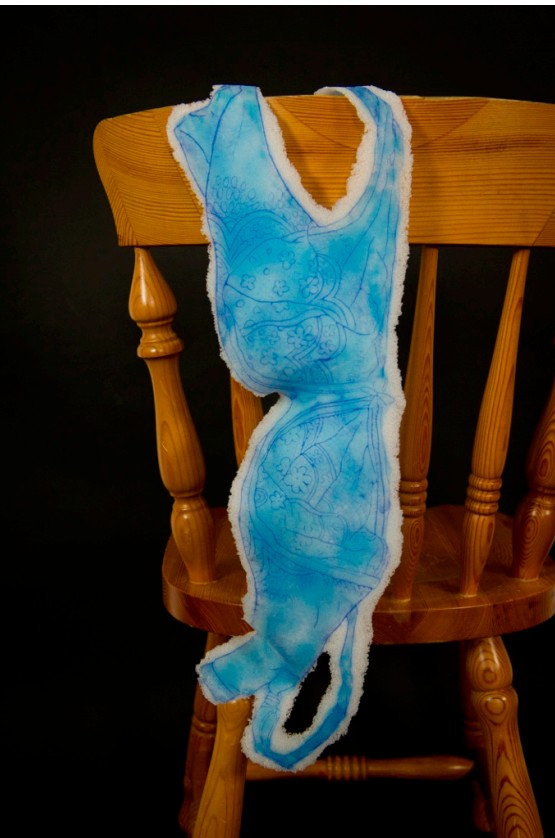

**Figure 21.** Release, from the series Let's Hook Up, Sophie Longwill. 2019. Wooden chair, pâte de verre glass, hot sculpted and slumped. Dimensions: variable. As shown on display in Northlands Creative, Scotland. [Created by author].

The process of making this series was, and continues to be, challenging for the artist. Researching the data on abortion, sexual assault, and gender-based violence, and the history of the infractions against women at the hands of the Irish state and church, such as the Magdalene laundries, was often difficult to process. It also triggered reflection on personal experiences of sexual assault and rape, and the artist realized that making these pieces was allowing her to finally speak about those experiences and come to terms with them. She reflected on the vulnerability of being an artist as she literally showed people her knickers and revealed secrets that had long been kept hidden. The more she worked on the artwork and spoke about the work with others, the easier and lighter it became as the shame slipped away, and a new powerful interaction came about—as she opened up about her personal experiences, the women around her, including family, friends, and complete strangers, came to join with their stories. It was heartbreaking to realize that this too, carrying the burden of these experiences, was a part of everyday life as a woman for so many. However, it was heartening to experience the connections formed, the tears shed, and the weight lifted. These pieces became a way to start the conversation. That is why it is important to the artist that the sculptures are not obviously crude or provocative in a way that would be off-putting, instead they have a vulnerable beauty that draws you in and can be taken at face value as simply a depiction of a garment or investigated and considered more deeply. The use of glass as the material is vital—it gives a feeling of ephemerality. It emphasizes the sense of delicacy and intimacy and the potential breaking of taboo.

## 7. Moving Forward

The referendum in Ireland did result in the repeal of the 8th Amendment. Abortion is now permitted in Ireland during the first twelve weeks of pregnancy, and later in cases where the pregnant woman's life or health is at risk, or in cases of fatal fetal abnormality. Unfortunately, that is not the case worldwide and, even when written into law, it is not guaranteed to last, as has become painfully apparent with the overturn of Roe v Wade in the USA last year. Women's rights to bodily autonomy and access to healthcare is a fragile issue across the world. The United Nations has 17 goals for sustainable development, of which one is to achieve gender equality and empower all women and girls. They state that 'gender equality is not only a fundamental human right, but a necessary foundation for a peaceful, prosperous, and sustainable world.' (UN 2023).

One of the most devastating obstacles facing women globally is the threat of sexual violence. One in three women worldwide experience physical or sexual violence, mostly by an intimate partner. Violence against women and girls is a human rights violation, and the immediate and long-term physical, sexual, and mental consequences for women and girls can be devastating, including death. Only 40 per cent of women seek help of any sort after experiencing violence (UN Women 2023).

## 8. Conclusions

It is clear that women's dress has always been closely linked to greater social issues. The evolution of female undergarments not only liberated women's bodies but also gave women more independence and provided employment in the manufacturing and sale of underwear. This essay has demonstrated how the development of underwear specifically for sex and seduction, i.e., lingerie, further emphasized the fetishization of the garment, which in turn was accentuated by lingerie advertising. Intimate apparel gained an anthropomorphic quality and came to stand in for the female body as well as being closely associated with a woman's propriety and morals. When underwear is reimagined in unexpected materials like glass, it creates a whole new set of stories. It can be beautiful, subversive, or funny. It becomes clear that the body and clothes can be read like texts, and using female underwear can represent not only women but the whole vast expanse of the female experience. As an everyday familiar object, referencing underwear provides a way for artists to engage an audience in difficult discussions and to spark ideas. Female underwear has such a rich and variegated history; it has built up layers of ambiguous

meanings and associations over time. Female underwear functions in contemporary art to provide a rich seam to mine and to tell stories, provoke, and inspire. It is Longwill's goal to continue developing and exhibiting this series in the hope that in some small way it can help to create an engaging safe space to begin these conversations about difficult topics, empowering women to connect and unburden.

**Funding:** This research received no external funding.

**Data Availability Statement:** Data is contained within the article. The data presented in this study are available in the following references.

**Conflicts of Interest:** The author declares no conflict of interest.

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
