# Peer review of "Knickers in a Twist: Confronting Sexual Inequality through Art and Glass"

_arts, 2023_

Round 1

Reviewer 1 Report

A well-researched and well-written account of the history of abortion in Ireland followed by the history of underwear leads intro a short description of the development of artwork in glass and culminates with an account of the author's bravery in openly referencing a past sexual assault, which leads to discussion with others with the same experience. I was astonished to see the statistics  related to sexual assault/violence. 

The title of the article is a touch heavy-handed, and I was slightly confused by the fact that there was very little about sexual assault/violence in the article -  after the introduction, it is only mentioned in the section about the artworks and in the statistics in moving forward. A different title, such as "confronting sexual inequality through art glass", might bring the different narratives together. Of course it wouldn't have the shock value.

Author Response

Thank you so much for the feedback, I agree and have changed the title. I also added about the research methodology in the introduction

Reviewer 2 Report

The paper focuses an important topic, it is well written, and it is biographic. Emphases on current issues and is very sensitive in explaining them. The way the author writes makes it interesting when reading. However, there is a lack of academic writing and methodology in the text. 

a)    All figures should be referred to in the text. Also, figures should appear after been mentioned on the text. figure 1 is not contextualised.

b)    In Point 4 – the art of knicker line  233 - Needs further development and theoretical depth in the artists presented.

c)    As the artist is using glass as a chosen material for the art pieces, it would be important to see if this already happen in the glass art. Is there other artists that are also working in these theme?

d)    In the abstract is written – “The author will detail the artistic processes involved in making the work as well as the conceptual development, exploring material and meaning” . This part should also be more developed. There are only 2 images of the author work, please provide more. Also it would be important to have photos of the process….

Line 42 – reference for “the 8th amendment was a subsection introduced to the Constitution of Ireland in 1983"

Line 67 - when IMELDA started?

In point 3 line 84

This is the historical part, so there should be more references to support what are you saying, especially when dates are mentioned.

Line 85 – you should have a reference

Line 87-89 – In this sentence you are making an assumption. It is a personal opinion? it would be important to have examples of comparison.

Line122 – reference?

Line 175 – please change “world war 1” to “world war I”

Line 214 - “ many feminist artist have made”, please give examples

Line 217 – “ feminist artists”…. Who? give examples

In Point 4 – the art of knicker line  233

Needs further development and theoretical depth in the artists presented.

Artists are mentioned such as Sarah Lucas, VALIE EXPORT, Scout Pare-Philips and Allen Hatstand.

A specific works was chosen in the figure 2, but never mentioned in the text. It would be important to develop more this point 4. Please add a work on the text of the artist, speak about the work and use references for each artist.

For a better methodological approach the author can have :

4.1 – artists …… (and have the four artists mention) ;

4.2 - knicker in glass /or glass sex cloths /or paradigm for womanhood (glass artists);.......

4.3 - author work……

Line 261 – stained glass is more than a window in a catholic church. Nowadays the perception change, the context, and the place. It also could be very provocative to have it in a stained glass due to all the context of the past…..

Line 264 – “but they were had a cartoonish quality that felt too playful for the underlying context. “ – confusing… 

It would be interesting if you add figures of the process.

Line 268 – “patterns to replicate textile “ – where the ideas of the patterns come from?

Line 287-288 –“while also experimenting with their display in exhibition settings to make the work increasingly dynamic and engaging”. This is very interesting. Please develop more. Why was important to have the pieces display in the exhibition? How was the public reaction? Did you have any contact from IMELDA about the work you did?

Please provide more photos of the work.

Reviewer 3 Report

This manuscript explores the “Knickers in a twist: Confronting Sexual Assault through Art  Glass”. The manuscript is elaborately described and contextualized with the help of previous and present theoretical background. All the references cited are relevant to this area of research.  The conclusion are supported by the present studies . The manuscript needs the following modifications before the acceptance.

1. Abstract: The Sentence “Knickers. Big, small, plain, sensual, provocative or practical” is not completed

1. Abstract – Mention your research recommendations.

2. Arrange the key words in alphabetical order. Also use sentence case

4. Do not use words like I, We, You. My etc.

5. What is the novelty of your work?

9. Increase the number of references.

Minor editing of English language required

Round 2

Reviewer 2 Report

The article is very well strategized. I would like to congratulate the author on the effort made. Is an up-to-date article that demonstrates meticulous and careful research with a pertinent study of the art. The selected artists are relevant and are consistent with the Sophie Longwill work.

To improve the article.

a)     In Sophie Longwill pieces the author explains the technique used. It can be interesting to explain the technique in the other figure’s legend, special the images where glass was used.

b)    Also, figure1 legend has the size of the piece, this is important. The other legends of  Sophie Longwill pieces should also have the dimensions.

c)     Line 504 – “These blown glass knickers by American artist Kati “. It is confusing. The pieces where never presented before, so the reader does not know what the author is speaking. Please change the sentence

d) Figure 15 and 20 are not mention in the text

Author Response

Have added more information and dimensions to all figures and fixed details relating to mentioned figures
